# Estimated Choline Intakes and Dietary Sources of Choline in Pregnant Australian Women

**DOI:** 10.3390/nu14183819

**Published:** 2022-09-16

**Authors:** Yasmine Probst, Dian C. Sulistyoningrum, Merryn J. Netting, Jacqueline F. Gould, Simon Wood, Maria Makrides, Karen P. Best, Tim J. Green

**Affiliations:** 1School of Medical, Indigenous and Health Sciences, University of Wollongong, Wollongong, NSW 2522, Australia; 2Illawarra Health and Medical Research Institute, Wollongong, NSW 2522, Australia; 3Discipline of Paediatrics, Adelaide Medical School, Faculty of Health and Medical Sciences, University of Adelaide, Adelaide, SA 5000, Australia; 4Women and Kids Theme, South Australian Health and Medical Research Institute, Adelaide, SA 5000, Australia; 5Factors Group of Nutritional Products Inc. Research & Development, Burnaby, BC V3K 6Y2, Canada; 6School of Public Health, Faculty of Health Sciences, Curtin University, Perth, WA 6102, Australia

**Keywords:** choline, intakes, diet sources, pregnancy, Australia

## Abstract

(1) Background: Despite the postulated importance of choline during pregnancy, little is known about the choline intake of Australians during pregnancy. In this study, we estimated dietary intakes of choline in early and late pregnancy, compared those intakes to recommendations, and investigated food sources of choline in a group of pregnant women in Australia. (2) Methods: 103 pregnant women enrolled in a randomized controlled trial. In early pregnancy (12–16 weeks gestation) and late pregnancy (36 weeks gestation), women completed a food frequency questionnaire designed to assess dietary intake over the previous month. (3) Results: Choline intakes and sources were similar in early and late pregnancy. Median choline intake in early pregnancy was 362 mg/day. Of the women, 39% and 25% had choline intakes above the Australian National Health and Medical Research Council (NHMRC) adequate intake (AI) of >440 mg/day and the European Food Safety Authority (EFSA) AI of >480 mg/day for choline in pregnancy, respectively. Eggs, red meat, nuts, legumes, and dairy accounted for 50% of choline intake, with eggs being the most significant contributor at 17%. (4) Conclusions: Few pregnant women in our study met the AI recommended by the NHMRC and EFSA. In Australia, choline intake in pregnancy may need to be improved, but further work to define choline requirements in pregnancy is required.

## 1. Introduction

Choline is an essential nutrient required for synthesizing the neurotransmitter acetylcholine and the methyl group donor betaine; it is a component of phospholipids [1]. Choline can be synthesized from phosphatidylcholine in the body, but endogenous synthesis may be insufficient to meet requirements, especially during periods of rapid growth [2]. During pregnancy, choline requirements increase as the fetus requires large amounts of choline for brain development [3]. A lack of choline during pregnancy has been associated with poorer cognitive outcomes in cohorts [4]. However, high-quality randomized controlled trial data are lacking.

In 2005, the Australian National Health and Medical Research Council (NHMRC) adapted the 1997 U.S. Institute of Medicine adequate intakes (AI) for choline [5,6]. For pregnancy, this was set at 10 mg/day lower than the US at 440 mg/day. Based on more recent evidence, the European Food Safety Organization (EFSA) set the AI at 480 mg/day for pregnancy in 2016 [7]. Most surveys during pregnancy suggest that intakes are well below the AI [8]. The only Australian data available were collected during the National Nutrition and Physical Activity Survey (NNPAS) 2011–2012, and indicates that less than 1% of pregnant women had choline intakes exceeding the NHMRC AI [9,10]. However, this survey was conducted over ten years ago, so its findings were not compared against the newer EFSA recommendations for choline, nor did it consider food sources of choline during pregnancy [7].

As part of a randomized controlled trial (RCT) of folic acid supplementation, we obtained estimates of choline intakes using a food-frequency questionnaire at study entry (12–16 weeks) and ~36 weeks’ gestation [11]. Here we report the dietary choline intakes and related nutrients in early and late pregnancy, compare choline intakes to recommendations, and explore food sources of choline in a group of Australian pregnant women.

## 2. Materials and Methods

The data described here were collected as part of an RCT. Details of the study design have been described previously [11]; here, we highlight aspects of the study methods relevant to the choline sub-study.

### 2.1. Study Participants and Procedures

The term “pregnant women” in this study refers to any person with the potential to be pregnant, including trans men and non-binary people [12]. To be eligible, women had to live in Adelaide, South Australia, be between 12–16 weeks gestation carrying a singleton pregnancy, currently taking a folic acid supplement, and planning to continue taking a folic acid-containing supplement throughout pregnancy. Participants were recruited in person at their first antenatal appointment or through a trial recruitment company (TrialFacts, Melbourne, Australia) utilizing an online digital marketing campaign. Participants were randomized to a prenatal micronutrient supplement with (800 µg) or without (0 µg) folic acid. The supplement did not contain choline. An online self-administered food frequency questionnaire was completed at study entry (early pregnancy; 12–16 weeks) and late pregnancy (34–36 weeks). At study entry, women also reported age, income, and pre-pregnancy body mass index (BMI). All participants provided informed consent, and the study was approved by the Women’s and Children’s Health Network Research Ethics Committee—HREC/19/WCHN/018 and Flinders Medical Centre—SSA/20/SAC/61.

### 2.2. Choline Intakes

Choline intakes were determined using the online version of the Dietary Questionnaire for Epidemiological Studies (DQES v3.2) developed by the Cancer Council of Victoria [13]. The DQES v3.2 is an 80-item semiquantitative food-frequency questionnaire (FFQ). It has undergone validation in several populations, including various ethnic backgrounds, including young adults [14], women of childbearing age [15], and people living with diabetes [16]. The DQES was initially developed to assess dietary intake over the previous year but was modified to assess intake over the prior month. In a study of young adults, the DQES showed acceptable validity and reproducibility over one month [14]. As the DQES 3.2 does not include data for choline, we used choline food-composition values recently added to the AUSNUT 2011–2013 database by Probst et al. [9]. In brief, the choline data was created from a systematic literature search of published studies and food composition data. The data were matched to AUSNUT food codes; foods not specified in detail and composite items were informed by the Food Standards Australia New Zealand (FSANZ) recipe files [17]. The DQES includes proprietary information for determining the nutrient values, and choline values were assigned to each of the 80 DQES food items. Choline values previously matched to food items from the AUSNUT 2011–2013 database were considered. All relevant items from AUSNUT were mapped to the DQES food items. For example, the DQES item ‘full cream milk’ was mapped to 15 ‘full cream milk’ food items in AUSNUT. Food entries not specified were excluded due to the use of mean data to determine these values. Box plots were created to confirm the relevance of all mapped foods to the food items, with outliers’ food items considered individually for relevancy and, if necessary, excluded (11%).

### 2.3. Statistics

Descriptive statistics are reported as means ± SD, median (IQR), *N* (%), and % (95% CI) as appropriate. Statistical analyses were completed using SPSS 28.0 (I.B.M. Corp., Armonk, NY, USA).

## 3. Results

### 3.1. Participants

Pregnant women were recruited and enrolled in the trial between December 2019 and June 2020. Of the 639 women assessed for eligibility, 103 were enrolled and randomized. Of the 103 participants, 93 completed the DQES FFQ at early pregnancy (12–16 weeks), and 84 completed the FFQ at late pregnancy (34–36 weeks). Eighty women completed both FFQs. Mean gestational age ± SD at study entry was 13.2 ± 1.2 weeks. The mean maternal age was 31 years, and more than 85% of the participants were of European ethnicity. More than 87% of participants had completed secondary education, and 60% had an annual household income higher than AUD 105,001 (Table 1). Of the participants, 55% had previously given birth to one or more children. Most participants had a BMI in the healthy range.

### 3.2. Choline Intakes

Median choline intakes were similar in early and late pregnancy, 394 and 418 mg/day, respectively (Table 2). In early pregnancy, 39% of participants exceeded the National Health and Medical Research Council’s (NHMRC) AI for choline during pregnancy (>440 mg/day), which rose to 51% by late pregnancy. Using The European Food Safety Authority’s (EFSA) higher AI (>480 mg/day) for choline, only 25% and 33% of participants exceeded the AI in early and late pregnancy, respectively.

### 3.3. Sources of Choline

Food sources of choline were similar for early and late pregnancy. Food or food groups ranked by their contribution to daily choline intakes during pregnancy are shown in Table 3. Eggs were the most significant contributor to choline intake at both time points, providing 72–76 mg/day and approximately 17% of total choline intake. Eggs, red meat, nuts and legumes, dairy, vegetables, and chicken accounted for around 70% of choline intake. The remaining food or food groups account for 5% of choline intake or less.

## 4. Discussion

Between 50–70% of pregnant participants in our study had choline intakes below NHMRC and EFSA recommendations. Choline intake was similar in early and late pregnancy, averaging 401 mg/day over both periods. More participants exceeded the NHMRC AI in late (51%) than early (39%) pregnancy. Indeed, the lowest intakes published [8] were in pregnant women in the Australian NNPAS 2011–2012 [9], where the mean choline intake was only 251 mg/day, and <1% of women met the NHMRC AI.

There are several reasons why the choline intakes were higher in our study than in NNPAS 2011–2012. We used an FFQ which assessed usual nutrient intakes over the month prior, whereas the method used in NNPAS 2011–2012 relied on 24-h recalls on two different days per individual. Compared to FFQs, 24-h recalls may underestimate intakes of infrequently consumed foods such as eggs, which may be eaten once or twice per week and contain high amounts of choline [18]. Conversely, the FFQ, due to its long lists of foods, may overestimate the consumption of food items because there is a tendency to report a specific food as being eaten more than it has [18]. In the NNPAS 2011–2012, 24-h-recall energy intakes were under-reported by 22% of subjects (NNPAS underestimation), leading to all nutrients being under-estimated.

Furthermore, a higher pre-pregnancy BMI is also known to increase the under-reporting of energy intake, leading to the under-reporting of other nutrients, including choline, during the later stages of pregnancy [19]. Pre-pregnancy BMIs were not given in the NNPAS; however, the BMIs of non-pregnant participants of childbearing age were higher than pre-pregnancy BMIs in our study. The phenomenon of under-reporting, known to occur with 24-h recalls, may have been counteracted by the overestimation of energy and nutrient intake inherent with FFQs [18], though this requires further confirmation.

Derbyshire et al. have recently published a comprehensive review of ’habitual’ choline intakes across childbearing years [8]. Although the choline intakes observed in our study were higher than those reported in others in this review, globally, there is considerable variation in reported intake. In the USA, the NHANES mean intake was ~320 mg/day despite using 24-h recalls [20]. In contrast, studies using an FFQ, in Vancouver (Canada) and Eastern Massachusetts reported mean intakes for pregnant participants of 383 and 344 mg/day, respectively [21,22]. All studies have reported considerable variations in choline intake within a population; in many cases, the standard deviation is larger than the mean. This variation is not surprising given that choline is found in high quantities in certain foods that may or may not be eaten during pregnancy, such as eggs or fish.

In the NNPAS 2011–2012, as in our study, eggs were the highest contributor (9%) to choline intakes for all Australian population groups [9]. However, the contribution of eggs to the choline intake of Australian pregnant women was not detailed. Most other studies of choline intake in women of childbearing age have not reported food sources of choline. One exception is the Alberta Pregnancy Outcomes and Nutrition (APrON) cohort, where the authors reported that dairy was the number one source of choline at 21%, followed by eggs at 12% [23].

A limitation of our study was the small sample size and the use of a cohort from an existing randomized controlled trial. However, our participants were not dissimilar to the broader population of Australian pregnant women. In 2019, the mean age of first-time mothers in Australia was 29.4 years, slightly lower than the mean age of 31.1 years in our study [24]. However, over 50% of our participants had given birth to at least one previous child. Of our participants, 83% described themselves as of European ethnicity; data showing maternal country of birth suggest that this is higher than the national average [24]. Sixty percent of participants had an annual family income of over AUD 105,000, similar to the median average Australian family income (2020) of AUD 120,000 [25]. A second limitation of our study is that choline only has an AI, not an estimated average requirement. When a population exceeds an AI, we can be confident that the population is receiving an adequate amount of the nutrient; however, being below an AI does not mean the population is deficient.

In conclusion, few pregnant women in our study met the AI recommended by the NHMRC and EFSA. Choline intake in Australia may need to be improved; however, more data are required on the clinical consequences of inadequate choline intake during pregnancy.

## Figures and Tables

**Table 1 nutrients-14-03819-t001:** Characteristics of participants at study entry.

Characteristic	Mean SD or *N* (%) ^1^
Age (years)	31.1 ± 4.8
Gestational age at study entry	
12–<14 weeks	64 (62)
≥14–16 weeks	39 (38)
Pre-pregnancy BMI (kg/m^2^) ^2^	24.1 ± 4.7
Healthy (18.5–24.9)	68 (72)
Overweight (25.0–29.9)	12 (13)
Obese (30.0 and above)	14 (15)
European ethnicity	85 (83)
Completed secondary education	90 (87)
Annual household income (AUD)	
$70,000 or less	18 (17)
$70,001–$105,000	19 (18)
$>105,001	61 (60)
Prefer not to disclose	5 (5)
Parity	
0	47 (46)
1	45 (44)
>1	11 (11)

^1^ *N* = 103. ^2^ Body Mass Index (*N* = 94).

**Table 2 nutrients-14-03819-t002:** Median choline intake and percentage of women consuming less than the National Health and Medical Research Council (NHMRC) and European Food Safety Authority (EFSA) adequate intake (AI) for choline in early and late pregnancy.

	Early Pregnancy (*N* = 93)	Late Pregnancy (*N* = 84)
Choline intake (mg/day)	362	414
Interquartile range	298–484	(303–509)
<NHMRC AI ^1^ % (95% CI)	61 (51, 70)	49 (39, 59)
<EFSA AI % (95% CI)	75 (65, 84)	67 (56, 77)

^1^ 440 mg/day choline.

**Table 3 nutrients-14-03819-t003:** Choline intake by reported DQES food or food group in early and late pregnancy ^1^.

Food or Food Group	Early Pregnancy (*N* = 93)		Late Pregnancy (*N* = 84)	
	mg/Day	% Intake ^2^	mg/Day	% Intake ^2^
Eggs	72 ± 55	17 ± 11	76 ± 59	17 ± 10
Red Meat	49 ± 39	12 ± 11	58 ± 42	14 ± 9
Nuts and legumes	46 ± 58	11 ± 13	50 ± 61	11 ± 12
Dairy	40 ± 29	10 ± 7	43 ± 27	11 ± 6
Vegetables	35 ± 22	9 ± 5	30 ± 20	8 ± 5
Chicken	30 ± 19	8 ± 6	29 ± 19	8 ± 5
Fruit	20 ± 13	5 ± 4	19 ± 14	5 ± 4
Pasta	18 ± 11	5 ± 3	19 ± 12	5 ± 3
Fish	16 ± 16	4 ± 4	18 ± 16	4 ± 4
Bread	12 ± 8	3 ± 3	12 ± 7	3 ± 3
Breakfast cereals	12 ± 10	3 ± 3	13 ± 15	3 ± 3
Beverages	10 ± 14	2 ± 3	13 ± 12	3 ± 3
Other savory foods	10 ± 6	3 ± 2	8 ± 5	2 ± 1
Processed meat	7 ± 7	2 ± 2	9 ± 8	2 ± 2

^1^ Values are Mean ± SD. ^2^ Mean daily percentage of choline from food or food group. DQES: Dietary Questionnaire for Epidemiological Studies vs. 3.2.

## Data Availability

The data presented in this study are available on request from the corresponding author.

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
