# Peer review of "Estimated Choline Intakes and Dietary Sources of Choline in Pregnant Australian Women"

_nutrients, 2022, doi:10.3390/nu14183819_

Round 1
Reviewer 1 Report
This is a very nice study and a well written paper that requires few revisions. The title and/or abstract should reflect that this is an estimate of choline intake as described in Methods. There is no introduction or discussion of the other nutrients reported in Table 2. The sentence on line 66-67 is incomplete.
Author Response
We thank the reviewer for their kind comments.
Reviewer 1 Comment 1: Title and/or abstract should reflect that this is an estimate of choline intake as described in the methods
Response 1: We agree, and the title has been changed to: Estimated choline intakes and dietary sources of choline in pregnant Australian women. The word estimate has been added to the abstract at Line 20
Reviewer 1 Comment 2: There is no introduction or discussion of the other nutrients reported in Table 2. Response 2: Both reviewers have raised this; thus, we have removed data for the other nutrients. Reviewer 1 Comment 3: Conclusion needs improvement Response 3: We have changed the conclusion at Line 196:In conclusion, few Australian pregnant women in our study met the AI recommended by the NHMRC and EFSA. Choline intake may need to be improved; however, we need more data on the clinical consequences of inadequate choline intake during pregnancy.
Reviewer 2 Report
Dietary intake of choline and choline sources during Australian pregnancies
OVERALL
The DQES was administered twice during pregnancy: at 12-16 weeks and 34-36 weeks. What timeframe is the questionnaire meant to capture? Is there an overlap between the two time periods? The online version says the DQES captures the past 12 months, but the discussion section of this manuscript says the past 1 month. This timeframe should be explicit in the Materials and Methods.
What proportion of participants completed the DQES at both timepoints? It might be interesting for the authors to report the intraclass correlation coefficient from those with both timepoints, in order to get a better feel for variability over time.
Suggest the authors use mg/day not mg/d.
ABSTRACT
Methods: Can the authors expand on the methods by including the timeframe of the study (2019-2020) and when women were enrolled (12-16 weeks’ gestation). I also suggest a term other than “females” be used (woman is fine given the definition included in section 2.1)
Results: “based on” appears out of place in line 27.
Results: Include abbreviations for the NHMRC and EFSA upon first use
INTRODUCTION
Lines 40-41: Include citation for “A lack of choline during pregnancy 40 has been associated with poorer cognitive outcomes in cohorts”
Lines 47-48: Include citation for “Most surveys during pregnancy suggest that intakes are well below the AI.”
Lines 55-47: Missing “and” before “explore food sources of choline in a group of Australian pregnant females.”. I also suggest a term other than “females” be used (woman is fine given the definition included in section 2.1)
MATERIALS AND METHODS
Line 64: “use of the term” is extraneous in this sentence.
Lines 66-67: “Those with a singleton pregnancy between 12- and 16-weeks 66 gestation, taking a folic acid supplement, and planning to continue supplement use” is not a full sentence.
Lines 95-98: What number and proportion of items were not specified and thus excluded?
Line 98: Suggest the end of this sentence is fixed to “individually for relevance and, if necessary, excluded”. What number and proportion were removed?
Line 75 and Results, line 109-110: What was the significance of the blood draw in this analysis? No data from the venipuncture was used so why was this mentioned?
RESULTS
Table 2: Why were values for other nutrients included in this table? Explain their relevance to the study question and add details to the materials and methods, or remove them altogether. Were these only from the study supplement or were these values obtained from the DQES?
Table 3: Suggest the authors keep the food groups on the same line rather than list according to intake in early and late pregnancy. For example, the “Bread” row in early pregnancy becomes “Breakfast cereals” in late pregnancy, which is more difficult to read than having the entire row be the values for “Bread”.
DISCUSSION
Lines 158-159: A higher pre-pregnancy BMI increases under-reporting of what exactly? Choline? Energy intake?
Lines 161-163: “It may be assumed that the reporting bias of the participants may have balanced the methodological over-estimation seen with an FFQ, though this does require further confirmation.” This statement didn’t make sense to me. Does the FFQ methodologically overestimate? Or does a 24-hour questionnaire underestimate, making FFQs appear to overestimate? Nevertheless, the NNPAS relying on 24-hour questionnaires and having a higher average participant BMI may have contributed to underestimation in this cohort. I am not sure how this leads to a balance with the reporting bias in the current study though.
